# Comparison of Tandem Mass Spectrometry and the Fluorometric Method—Parallel Phenylalanine Measurement on a Large Fresh Sample Series and Implications for Newborn Screening for Phenylketonuria

**DOI:** 10.3390/ijms24032487

**Published:** 2023-01-27

**Authors:** Dasa Perko, Urh Groselj, Vanja Cuk, Ziga Iztok Remec, Mojca Zerjav Tansek, Ana Drole Torkar, Blaz Krhin, Ajda Bicek, Adrijana Oblak, Tadej Battelino, Barbka Repic Lampret

**Affiliations:** 1Clinical Institute for Special Laboratory Diagnostics, University Children’s Hospital, UMC Ljubljana, 1000 Ljubljana, Slovenia; 2Department of Endocrinology, Diabetes and Metabolic Diseases, University Children’s Hospital, UMC Ljubljana, 1000 Ljubljana, Slovenia; 3Faculty of Medicine, University of Ljubljana, 1000 Ljubljana, Slovenia; 4Department of Nuclear Medicine, UMC Ljubljana, 1000 Ljubljana, Slovenia

**Keywords:** tandem mass spectrometry (MS/MS), fluorometric method (FM), comparison, phenylalanine (Phe), phenylketonuria (PKU), newborn screening (NBS), false positive, recall rate

## Abstract

Phenylketonuria (PKU) was the first disease to be identified by the newborn screening (NBS) program. Currently, there are various methods for determining phenylalanine (Phe) values, with tandem mass spectrometry (MS/MS) being the most widely used method worldwide. We aimed to compare the MS/MS method with the fluorometric method (FM) for measuring Phe in the dried blood spot (DBS) and the efficacy of both methods in the NBS program. The FM was performed using a neonatal phenylalanine kit and a VICTOR2^TM^ D fluorometer. The MS/MS method was performed using a NeoBase^TM^ 2 kit and a Waters Xevo TQD mass spectrometer. The Phe values measured with the MS/MS method were compared to those determined by the FM. The cut-off value for the NBS program was set at 120 µmol/L for FM and 85 µmol/L for MS/MS. We analyzed 54,934 DBS. The measured Phe values varied from 12 to 664 µmol/L, with a median of 46 µmol/L for the MS/MS method and from 10 to 710 µmol/L, with a median of 70 µmol/L for the FM. The Bland–Altman analysis indicated a bias of −38.9% (−23.61 µmol/L) with an SD of 21.3% (13.89 µmol/L) when comparing the MS/MS method to the FM. The Phe value exceeded the cut-off in 187 samples measured with FM and 112 samples measured with MS/MS. The FM had 181 false positives, while the MS/MS method had 106 false positives. Our study showed that the MS/MS method gives lower results compared to the FM. Despite that, none of the true positives would be missed, and the number of false-positive results would be significantly lower compared to the FM.

## 1. Introduction

Advances in laboratory technology, such as tandem mass spectrometry (MS/MS), which is more rapid, specific, and sensitive than traditional assays, have expanded the number of inborn errors of metabolism (IEM) detectable through newborn screening (NBS) programs [1]. MS/MS enables simultaneous analysis of a variety of amino acids and acylcarnitines extracted from dried blood spots (DBS) to detect various aminoacidopathies, organic acidemias, and fatty acid oxidation disorders [2].

Phenylketonuria (PKU; OMIM 261600) is the most prevalent IEM caused by pathogenic variants in the phenylalanine hydroxylase gene (*PAH*, OMIM * 612349) [3], which catalyzes the hydroxylation of phenylalanine (Phe) to tyrosine (Tyr) [4]. Phe and its secondary metabolites accumulate in the blood and brain [5,6,7] and cause irreversible nerve cell damage if left undiagnosed and untreated [8,9,10].

The development of Robert Guthrie’s diagnostic test in the early 1960s allowed mass screening of elevated Phe values, enabling early diagnosis of PKU [6,8,9,10]. In Slovenia, the NBS program for PKU was established in 1979 [11]. For 13 years, screening was performed using the Guthrie test [10,11,12,13], a semiquantitative method with limited sensitivity [14]. In 1992, the Guthrie test was replaced with a more sensitive and precise quantitative fluorometric method (FM). With the implementation of MS/MS technology in 2018, more specific and sensitive Phe measurement was enabled. With this technology, the expansion of 17 additional metabolic disorders was also implemented in our laboratory [15]. 

PKU was the first disease identified by the NBS program; however, different methods are now used in different countries or regions globally [10], from enzymatic and bacterial inhibition assays to fluorometric methods and MS/MS method [16]. In Japan, over 5400,000 neonates have been screened for PKU using an enzymatic assay successfully for over twenty years [17]. Therefore, it is important to understand the clinical implications and consequences resulting from the use of different methods. 

The primary aim of this study was to compare the MS/MS method with the FM for measuring Phe in the DBS. In the second part of this study, we aimed to compare MS/MS and the FM for PKU screening.

## 2. Results

### 2.1. Methods Comparison

We analyzed 54,934 DBS from newborns. The Phe values, measured with the MS/MS method, were compared to those determined by the FM. The measured Phe values varied from 12 to 664 µmol/L, with a median of 46 µmol/L for the MS/MS method, and from 10 to 710 µmol/L, with a median of 70 µmol/L for the FM (Figure 1). Lower quartile (Q1) and uper quartile (Q3) were 41 and 52 µmol/L for the MS/MS method, and 60 and 80 µmol/L for the FM. The MS/MS method’s 99th percentile was 73 µmol/L and the FM’s 99th percentile was 110 µmol/L.

The Bland-Altman analysis was performed, and the percent differences in Phe values between the MS/MS method and the FM were plotted against their average values (Figure 2). The calculated bias was −38.9% (−23.61 µmol/L) with a standard deviaton (SD) of 21.3% (13.89 µmol/L), indicating that Phe values, measured with MS/MS, were 38.9% lower than values, measured with the FM.

The value of the Spearman correlation coefficient was 0.49, indicating a moderate correlation between the two methods (Figure 3).

### 2.2. Methods Comparison—Implication for NBS

We analyzed 54,934 DBS from newborns. In 187 samples, the Phe value exceeded 120 µmol/L using an FM and 22 of these samples were classified as NBS positive, necessitating further conformation analyses and investigation. Phe values of 22 NBS positive newborns, S-Phe, genotype, diet, and true/false positive determination are listed in Table 1. The Phe cut-off for the MS/MS method was set at 85 µmol/L using the 99.98th percentile. In 112 samples, the Phe value exceeded the cut-off.

Six NBS-positive patients required a low-Phe diet and thus were classified as true positives. A total of 181 newborns were classified as false positives using the FM, and 106 using the MS/MS method (Figure 4). Positive predictive values were 5.36% for the MS/MS method and 3.21% for the FM (Table 2).

#### Adjustment of the Cut-Off Value for the MS/MS Method

We simulated the number of recalls with different Phe cut-off values ranging from 85 to 300 µmol/L. At the cut-off of 85 µmol/L, the number of recalls would be 112, which is five-fold higher than at 110 µmol/L. From 110 to 290 µmol/L, the number of recalls would drop minimally, from 21 to 9, with the number of false negatives remaining at zero. At the cut-off of 300 µmol/L, the first false-negative results would be obtained (Figure 5).

## 3. Discussion

NBS is a system that identifies apparently healthy newborns with inherited disorders, most of them metabolic in origin, before they cause any serious morbidity or even death [1,13]. It was first introduced in 1963 in the USA with the Guthrie test for detecting PKU [18]. Later on, different NBS approaches were utilized, including chromatography, fluorometry, radioimmunoassay, and, in the last almost three decades, MS/MS [6,10].

The MS/MS method proved to be among the most significant developments in clinical diagnostics and signified a paradigm shift from the traditional “one test–one disorder” model to the “one test–many disorders” strategy [19,20]. This allowed the expansion of NBS programs to encompass screening for various diseases in a single run.

In many countries, MS/MS technology is now used to measure Phe values in DBS. Although this method offers better analytical sensitivity than the FM and the benefit of evaluating several analytes simultaneously, it is unavailable in the majority of developing countries [21,22]. Countries that screen for PKU only usually use the FM as opposed to the MS/MS method, as it is less costly for them. However, when screening for more IEMs, the MS/MS method is the sole option. MS/MS equipment is more costly; however, when more IEMs are screened for, the cost per disease decreases significantly [14]. It is vital to perform such comparisons because, even though the development of the NBS program is geared toward the use of MS/MS, there are still many nations that employ the FM.

This study’s primary objective was to compare the FM and the MS/MS method for quantifying Phe in the DBS. Furthermore, both methods were compared in terms of PKU screening.

With the prolonged simultaneous use of the FM for PKU screening related to the process of expansion of NBS in 2018 with the MS/MS method, we had the opportunity to compare the two approaches for measuring Phe in DBS. There have been very few previous studies published on inter-method comparisons [23,24,25,26,27]. This is the first research that we are aware of that compares the FM to the MS/MS method for quantifying Phe in the fresh DBS. Immediate simultaneous analysis with both techniques enabled us to prevent variations between methods that may have resulted from the degradation of analytes in DBS during storage.

We analyzed 54,934 DBS from newborns born in Slovenia between 2018 and 2021. Our results demonstrated significantly higher results (23.6 µmol/L; 38.9%) obtained by the FM as compared to the MS/MS method.

There have been a few previous comparisons between the MS/MS method and amino acid analysis (AAA), and it has been demonstrated that the MS/MS method produces lower values than the AAA method does [23,24,25,26,27]. A recent study compared Phe values in DBS using the FM with a commercial kit and Phe values in plasma using the HPLC method. The result also showed higher values (9.8%) using the FM [22].

Among the most critical aspects of selecting a screening method is its sensitivity, which means that the number of false-negative results is kept to a minimum, ensuring that no patients are missed [12]. Loss of PAH activity results in a rise in Phe concentration in the body and toxic levels in the brain. Without treatment, PKU causes an increasing and persistent intellectual disability, as well as behavioral difficulties, seizures, delayed development, and mental symptoms [6,8,28]. Furthermore, another critical component, particularly from an ethical standpoint, is method specificity and a low rate of false positives. The definition of true or false positive was based solely on whether the patient required a low-Phe diet.

The FM has a cut-off value set at 120 µmol/L. This value was set in 1992 with the introduction of this method; for the determination, a pilot study with approximately 7000 NBS samples was conducted and the cut-off was set at the 99.98th percentile.

Our study has shown, that although the MS/MS method yielded lower measured values than the FM, none of the six true positives would be missed at the equivalent cut-off. The central finding of the second part of this study was that the number of false-positive samples is higher when using the FM than when using MS/MS. A total of 187 samples were detected as borderline or positive with the FM and 112 with the MS/MS method. The number of true positive patients was six. Thus, the FM yielded 181 false-positive samples, and the MS/MS method 106. A false-positive result is defined as a borderline NBS result or positive NBS results that are eventually determined as normal. False-positive results disturb the first weeks of a newborn’s life and can give parents real concern and long-term emotional stress, especially during that sensitive time when the mother and baby should bond [29,30,31,32]. From the physician’s and laboratory analyst’s perspectives, following up on each such result involves re-testing the first sample, contacting the clinic for further samples, testing them, incurring extra costs, and then notifying the clinic of the second result [12,31].

Using the MS/MS method, we simulated the number of recalls for a range of Phe cut-off values between 85 and 300 µmol/L. Increasing the cut-off from 85 to 110 µmol/L would significantly reduce the number of recalls. If BH4 deficiencies are not being targeted, we demonstrated that any cut-off value between 110 and 200 µmol/L is specific and safe, detecting all true-positive results and only a few false-positive ones.

Using the equivalent cut-off for comparison, the MS/MS method is more precise than the FM; it produced 106 false positives, 3 of which had higher Phe levels owing to other factors, including parenteral feeding and gestational-alloimmune liver disease (GALD). Furthermore, the MS/MS method using in vitro diagnostics (IVD) enables testing for almost 60 analytes simultaneously, including Tyr and, consequently, the Phe/Tyr ratio, an important factor in screening and disease monitoring [33,34,35,36,37].

## 4. Materials and Methods

### 4.1. Subjects

We analyzed DBS from 54,934 newborns born in Slovenia from 2018 to 2021. The register is likewise mandated by national legislation [2] and for inclusion in the NBS program or the IEM registry, families do not need to provide written informed consent. The study protocol related to genetic analyses of HFA patients was approved by the Slovene Medical Ethics Committee (No. 115/08/14). Written informed consent was obtained from all the participants or their parents before genetic analysis was performed [4].

### 4.2. Specimen Collection

Blood samples for NBS were taken between 48 and 72 h after birth from the newborn’s heel or by venipuncture and collected on blood collection paper (Whatman 903, LKB, Austria). Blood spots were allowed to dry at room temperature for at least 4 h before the analysis. All the samples were sent to the University Medical Center (UMC), Ljubljana. Half of the sample card was used at the Department of Nuclear Medicine (DNM) and the other half was sent to the Clinical Institute of Special Laboratory Diagnostics (CISLD), University Children’s Hospital Ljubljana.

### 4.3. The Fluorometric Method 

Measurements of Phe in DBS using an FM were performed at the DNM as part of a routine NBS, following an established diagnostic algorithm [38]. Each DBS sample was analyzed in duplicate using a neonatal phenylalanine kit (PerkinElmer Life and Analytical Sciences, Wallac Oy, Turku, Finland). The method is based on the enhancement of the fluorescence of a phenylalanine-ninhydrin product using the dipeptide L-leucyl-L-alanine [38]. 

A three-millimeter disc was punched from the DBS of each sample to a 96-well microplate. Extraction was accomplished by adding 15 µL of zinc sulfate-based solution followed by intense shaking for 2 min and incubation without shaking for 45 min at room temperature. After incubation, 40 µL of deionized water was added to each well and the microplate was gently tapped. A volume of 25 µL of the well contents was transferred to another 96-well microplate, 50 µL of a ninhydrin reagent was added, and mixing of the well contents was achieved by tapping the microplate. After the incubation step at 60 °C for 35 min, 200 µL of copper reagent was added, followed by intense shaking for 2 min and incubation without shaking for 45 min at room temperature.

The VICTOR2^TM^ D fluorometer (Perkin Elmer, Waltham, MA, USA) was used to measure fluorescence. The method measures Phe quantitatively in the presence of other amino acids, using a 390 nm wavelength for excitation and a 486 nm wavelength for emission. All low and high positive DBS controls were added in every analytical batch and all results were evaluated with assay-dedicated software. The method is also included in the external quality control scheme (United Kingdom Newborn Screening External Quality Assessment Scheme, KNEQAS) for period evaluation of measured results and comparison to laboratories worldwide using the same or other methods for Phe determination in DBS.

### 4.4. MS/MS Method

Measurements of Phe in DBS using the MS/MS method were performed at the CISLD using a NeoBase^TM^ 2 Non-derivatized MSMS kit (PerkinElmer, Turku, Finland) A three-millimeter disc was punched from the DBS of each sample in a 96-well plate. Elution of Phe was accomplished by adding 125 µL of NeoBase 2 Extraction Solution containing NeoBase 2 internal standards and incubation for 20 min on a plate shaker set at 650 rpm and 45 °C. The elute was transferred to a new 96-well plate.

The Waters Xevo TQD (PerkinElmer, Turku, Finland) mass spectrometer, triple quadrupole coupled with an electrospray ionization source connected to an Aquity UPLC system, and Neolynx V 4.2 software were used for analyses. Ten µL of elute were injected using Neo MSMS Flow Solvent as a mobile phase. The initial isocratic flow was set at 0.200 mL/min. The gradient flow was adjusted as follows: 0.015 mL/min for 0.13 min, 1 mL/min for 1 min, 0.2 mL/min for 1.20 min, and 0.2 mL/min for 1.30 min. The ion source was operated in a positive mode. Multiple reaction monitoring mode was used with a mass transition of 166.1 > 120.1 for Phe and 172.1 > 126.1 for Phe internal standard. The cone and collision energy were set at 26 V and 14 V, respectively. Low- and high-DBS controls were added in every analytical batch to monitor the system’s precision and accuracy. The method is included in the external quality control scheme (ERNDIM, Special Assays in Dried Blood Spots) and the CDC Newborn Screening Quality Assurance Program).

### 4.5. Statistical Analysis 

The GraphPad Prism 8 (GraphPad Software Inc., San Diego, CA, USA) program was used to conduct all statistical analyses. The Bland–Altman analysis was used to compare the MS/MS method and the FM, the average bias, the relative bias, standard deviation (SD), and the relative SD (RSD) of the bias were calculated. Using the Spearman correlation coefficient, the relationship between the two methods was determined. Range, median, the first quartile (Q1), the third quartile (Q3), and the 99th and 99.98th percentiles were used to represent descriptive statistics. 

### 4.6. Implications for NBS for PKU

#### 4.6.1. Current Diagnostic Algorithm for NBS

The current cut-off value for Phe using the FM is 120 µmol/L [38]. A diagnostic algorithm following abnormal NBS results for PKU was already presented in our previous study [38]. Phe levels between 120 and 200 µmol/L were classified as borderline and were retested from the same DBS. Newly collected DBS were requested and analyzed in the case of Phe value above 120 µmol/L. All patients with Phe values above 120 µmol/L measured twice (i.e., the first and second DBS) or with Phe values greater than 200 µmol/L in the initial DBS were classified as NBS positive, requiring direct clinical referrals and confirmatory testing [38].

Confirmatory testing was previously described [7,38] and was performed by measuring serum Phe (S-Phe), using an in-house FM and genetic analysis of the *PAH* gene. In individuals where the S-Phe value exceeded 400 umol/L, a low-Phe diet was introduced. 

All NBS-positive individuals with Phe values under 400 umol/L for whom no dietary changes were needed were classified as false positives. Each patient who required a low-Phe diet was considered a true positive.

#### 4.6.2. Method Comparison—Implications for NBS 

After measuring all the samples, we set the Phe cut-off value for the MS/MS method using the same percentile as with the FM, which was the 99.98th percentile. The specificity, the true and false-positive rates, and positive predictive values for each method were compared.

#### 4.6.3. Adjustment of the Cut-Off Value—The MS/MS Method

Based on the data obtained from the phenotypic and genotypic analysis of 54,934 samples, we simulated adjusting the Phe cut-off value to different values to determine the most appropriate one. At the same time, we determined at what value a false-negative result occurs. After adjusting the Phe cut-off, we re-evaluated the number of false positives. 

All the patients were followed at the Department of Endocrinology, Diabetes, and Metabolic Diseases, UCHL.

## 5. Conclusions

Different methods for PKU screening are now used worldwide; thus, it is important to understand the clinical implications and consequences arising from the use of different methods. Our study revealed that measuring Phe in DBS with MS/MS yields 38.9% lower values compared to the FM. However, further research revealed that despite the lower measured values with MS/MS, no true positives would be missed at the equivalent cut-off. The number of false-positive results would be substantially lower than when using the FM, reducing the burden on families and on the health care system. We have also demonstrated that raising the cut-off to 110 µmol/L considerably reduces the number of recalls. We have shown that the MS/MS method is more appropriate for the NBS program for PKU. It is faster, more reliable, more precise, and more accurate.

## Figures and Tables

**Figure 1 ijms-24-02487-f001:**
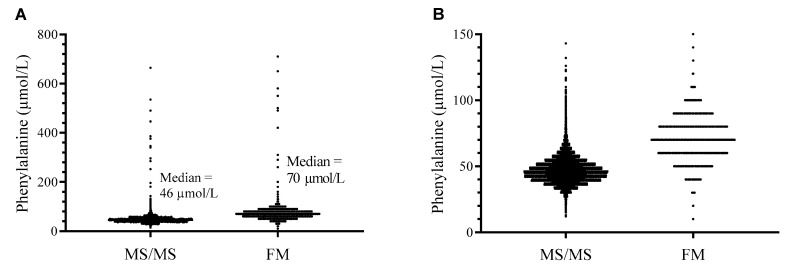
Scatter dot plot displaying the phenylalanine (Phe) values determined by the tandem mass spectrometry (MS/MS) and the fluorimetric method (FM). (**A**) The whole range of Phe values; (**B**) Phe values up to 150 µmol/L.

**Figure 2 ijms-24-02487-f002:**
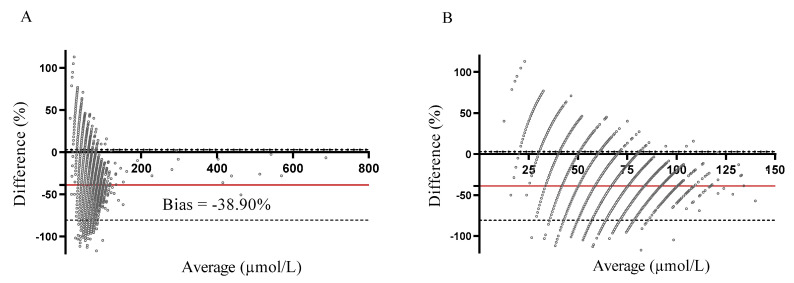
The Bland-Altman analysis of the tandem mass spectrometry (MS/MS) and the fluorimetric method (FM) for determining phenylalanine (Phe) concentrations in dried blood spots (DBS). The bias was −38.9%, with the lower (LLA) and upper (ULA) lines of agreement at −80.73% and 2.94%, respectively. (**A**) The whole range of Phe values; (**B**) Phe values up to 150 µmol/L.

**Figure 3 ijms-24-02487-f003:**
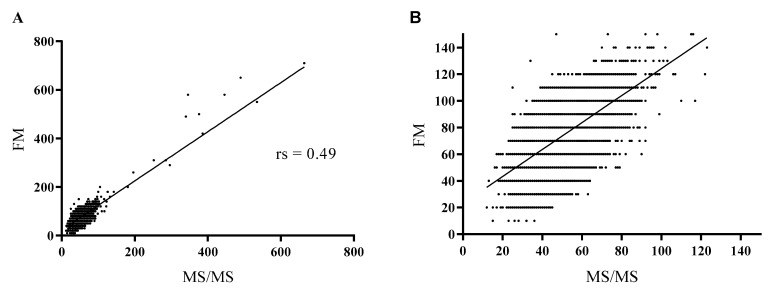
Correlation between the tandem mass spectrometry (MS/MS) and the fluorimetric method (FM). The Spearman correlation coefficient (rs) was 0.49, with a 95% confidence interval of 0.483–0.496. (**A**) The whole range of phenylalanine (Phe) values; (**B**) Phe values up to 150 µmol/L.

**Figure 4 ijms-24-02487-f004:**
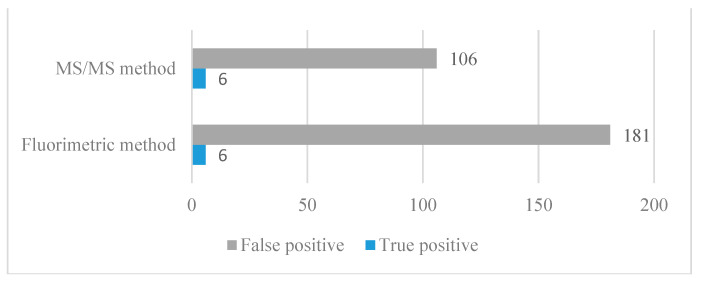
Comparison of the tandem mass spectrometry (MS/MS) and the fluorimetric method (FM) for phenylketonuria (PKU) newborn screening (NBS); the number of analyzed dried blood spots (DBS) was 54,934.

**Figure 5 ijms-24-02487-f005:**
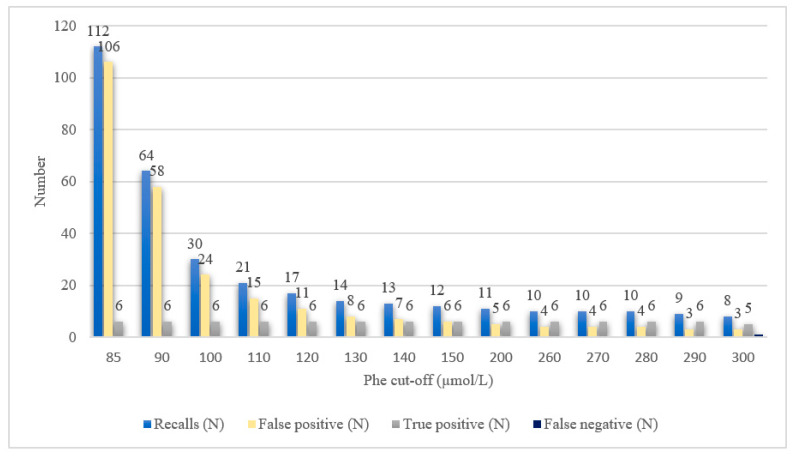
Simulated number of recalls, false-positive and false-negative samples at different phenylalanine cut-off values.

**Table 1 ijms-24-02487-t001:** Phe values, genotype, NBS classification, diet, and true/false positive definition of 22 neonates requiring conformational analysis.

Patient	DBS Phe (µmol/L)—Fluorometric Method	DBS Phe (µmol/L)—MS/MS Method	Recall DBS Phe (µmol/L)—Fluorometric Method	NBS +/−	Conformation Analyses and Diagnostic Algorithm			TP	FP	Note
					S-Phe (µmol/L)	Genotype (*PAH* Gene)	Diet			
1	500	376	/	+	569	NM_000277.1:c [143T > C(;)913-7A > G]	yes	X		
2	420	386	/	+	1118	NM_000277.2:c[143T > C(;)1222C > T]	yes	X		
3	550	535	/	+	802	NM_000277.2:c[842C > T(;)1222C > T]	yes	X		
4	490	340	/	+	751	NM_000277.3:c[473G > A];[473G > A]	yes	X		
5	580	446	/	+	983	NM_000277.3:c[473G > A];[473G > A]	yes	X		
6	290	296	/	+	593	NM_000277.1:c[442-5C > G(;)842C > T]	yes	X		
7	310	252	/	+	241	NM_000277.2:c[58C > T(;)165T > G]	no		X	
8	260	196	/	+	250	NM_000277.3:c[1208C > T(;)1222C > T]	no		X	
9	310	285	/	+	282	NM_000277.3:c[473G > A (;)1243G > A]	no		X	
10	180	143	200	+	229	NM_000277.3:c[473G > A(;)827T > C]	no		X	
11	160	132	180	+	200	NM_000277.3:c[898G > T(;)1208C > T]	no		X	
12	180	126	130	+	123	/	no		X	
13	150	116	120	+	247	NM_000277.3:c[678G > C(;)734T > C]	no		X	
14	150	115	170	+	194	/	no		X	
15	130	86	180	+	131	/	no		X	
16	130	74	120	+	92	/	no		X	
17	130	92	130	+	135	/	no		X	
18	180	100	160	+	144	/	no		X	
19	130	86	140	+	149	/	no		X	
20	650	490	/	+	67	/	no		*X*	*PN*
21	200	105	33	+	/	/	no		*X*	*↓GS*
22	710	664	/	+	417	/	no		*X*	*GALD*

DBS Phe—the phenylalanine (Phe) value in the dried blood spot (DBS); recall DBS Phe—the Phe value in newly collected DBS; S-Phe—the Phe value in serum; NBS +/− —newborn screening (NBS) positive or-negative result; diet—low-Phe diet; TP—true positive; FP—false positive; GALD—gestational-alloimmune liver disease.; PN—parenteral nutrition; ↓GS—low gestational age (under 32 weeks).

**Table 2 ijms-24-02487-t002:** Specificity, sensitivity, and, positive predictive values with two different methods.

	Recalls (N)	False Positive (N)	True Positive (N)	False Negative (N)	True Negative (N)	Sensitivity (%)	Specificity (%)	Positive Predictive Value (%)
Tandem mass spectrometry	112	106	6	0	54,822	100	99.81	5.36
Fluorometric method	187	181	6	0	54,747	100	99.67	3.21

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
