# Peer review of "Comparison of Tandem Mass Spectrometry and the Fluorometric Method—Parallel Phenylalanine Measurement on a Large Fresh Sample Series and Implications for Newborn Screening for Phenylketonuria"

_ijms, 2023, doi:10.3390/ijms24032487_

Round 1
Reviewer 1 Report
This is a nicely written paper comparing two methods of measuring phenylalanine in the screening for PKU. As you mention resource limited jurisdictions who want to screen for PKU will continue to use this and similar methods. There are some points around definitions and screening metrics which need attention as below
#2.6.1 this classification of screen positive is unusual – more accepted is a screen positive test is one requiring a further action on the baby hence would include both direct clinical referrals (over 200umol/L and requests for second samples). Please redo screening metrics using this definition of screen positive. Please also specify the criteria used to determine whether a diet was needed ie case definition. Note this doesn’t seem quite the same as the definition on p8l263.
Was there a difference in precision of the assays (inter- and intra-assay variability)? How accurate were the methods? What recovery of added phenylalanine was obtained? How did the methods compare on EQA?
#3.2.1 where there is an established screening method and comparison is done with a new method, and there is known bias between the methods as has been shown, the cutoff for the new method can be made the same as the old one by using the same centile or multiple of the median (120 umol/L is 2.61 MoM on TMS (median 46umol/L) and 1.71 on FM (median 70 umol/L)). Please redo the screening metrics using an equivalent cutoff for comparison with the metrics obtained using the same numeric cutoff.
Note also that a laboratory using MSMS is unlikely to be measuring only Phe and hence a more useful comparison of screening technologies is FM with MSMS incorporating phe/tyr ratio. Is it possible to use existing data to model this comparison (FM vs TMS Phe vs TMS Phe/Tyr)?
And some very minor mostly grammatical items
P2l70 please do not use filter paper – it isn’t filter paper, it is special blood collection paper.
P2l63 please be consistent with capitalisation or otherwise of chemical names (usually not)
P2l93 controls not previously described suggest start sentence at Controls ….
P3l96 which EQA scheme?
Reviewer 2 Report
This manuscript deals with the important topics of newborn screening (NBS) and phenylketonuria (PKU). The authors compare two frequently used methods for the determination of phenylalanine (Phe), tandem mass spectrometry (MS/MS) and a fluorimetric method (FM).
In my opinion, the main advantages of the manuscript are:
1. The two methods are compared contemporarily, i.e. both were run on the same fresh samples at the same time, instead of using stored samples as in some articles. Maybe this could be pointed out in the title too.
2. The sample size is large.
However, there are some issues with the design and implemetation of the study.
Major issues:
1. The authors have used the same cutoff for both methods. This raises questions as a cutoff from another method cannot be used elsewhere. Moreover, even if analyte levels of previously measured real patients are important, a cutoff cannot be set exclusively on the basis of these (a patient with a lower Phe level can come). The correct way to determine method-specific cutoffs in NBS is to measure several thousand (e.g. >5-10.000) real samples including some real patients and determine percentile ranges from the results. In NBS, 99,5-99,9-99,99 percentiles are often used. The cutoff should be updated regularly (e.g. every 1-3 years).
2. A cutoff of 120 uM was used for the FM method, determined in 1992. Again, cutoffs should be checked and corrected regularly. This is especially important if the laboratory sees that the number of false-positives is unacceptably high. While the reviewer agrees in that MS/MS can markedly decrease unnecessary recalls, a cutoff between 120-200 uM (as suggested for the MS/MS by the authors) would also have decreased the false-positive rate of the FM.
3. Was the diagnosis of true-positiveness based solely on the need for a diet? If yes, then what were the criteria for a diet initiation? Phe levels? Presence of certain genetic mutations? Clinical symptoms?
Minor issues:
1. L55 and L283 "NBS for PKU is considered a minimal current standard of care": This may be true in Europe but not globally. Other (especially developing) countries may have other preferences. Please rephrase.
2. L79: Why immediately duplicates? Generally, a single spot is analyzed first, and only in case of an abnormal result additional two from the same card are used.
3. L96 and L115: Does the laboratory participate in the CDC PT and QC schemes with one or the other method?
4. Table 1: Please rephrase "New DBS Phe" to "Recall DBS Phe" for clarity.
5. L215-222: It is generally true that if a country screens exclusively for PKU, the FM is generally cheaper. However, this paragraph in its currest state would suggest that FM is almost always cheaper than MS/MS which is not true. There are multiple factors to consider. MS/MS equipment is indeed more expensive, but its operation can easily be cheaper than FM if multiple disorders are screened for. This is especially true if laboratory developed tests (LDTs) are used. For many disorders there are no FM kits available. Please rephrase the paragraph to be a bit more comprehensive.
6. L236-238: This is not a real limitation: a) such high levels are probably higher than the upper limits of quantificaton (ULOQs), b) such high values generally come from real positives (provided that no parenteral
amino acid therapy was given), c) for screening purposes it does not really matter if the value is 600 or 700, both are positive.
7. L273-274 "using any cut-off value between 120 µmol/L and 200 µmol/L is safe": May be true only if BH4-deficiencies are not targeted. Please add.
8. L281: The Phe/Tyr ratio is important not just for monitoring, but also in screening. Please add.
Reviewer 3 Report
The tandem mass spectrometer is a very well-matched device for screening for various diseases, and we expect it to be widely introduced for newborn screening in the near future.
Author Response
Thank you for your comment.
Round 2
Reviewer 2 Report
The manuscript has been improved by incorporation of the reviewers' suggestions, including those of the present reviewer, thank you. I recommend accepting the manuscript in its present revised form.
Author Response
Thank you for your feedback and recommendation to accept the manuscript.